# Host immune players and their response to Hepatitis C therapies

**Kehkshan Jabeen**[1,2], **Madiha Khlaid**[3], **Sajid Mansoor**[4], **Ali Zalan**[3], **Momina Ejaz**[3], **Atika Mansoor**[5], **Aneela Javed**[3]*

1 Genomics Research Lab, Department of Biological Sciences, International Islamic University Islamabad, Islamabad, Pakistan, 2 Rawalpindi Medical University, Rawalpindi, Pakistan, 3 Department of Healthcare Biotechnology, Atta-ur-Rahman School of Applied Biosciences (ASAB), National University of Sciences and Technology (NUST), Islamabad, Pakistan, 4 University of Central Punjab (UCP), Lahore, Punjab, Pakistan, 5 Institute of Biomedical Genetic Engineering (IBGE), Islamabad, Pakistan

* javedaneela19@gmail.com

## Abstract

This study aimed to investigate alterations in the expression of four key cytokines (IL-7, IL-11, IL-15, and IL-27) and assess differential FAM26F expression in response to Hepatitis C virus (HCV) infection. Additionally, it sought to analyze changes in these cytokines after treatment in 244 chronic HCV patients and 28 controls undergoing various treatments, including standard interferon, pegylated interferon, and Direct Acting Antivirals (DAAs). The objective was to compare immune system regulation between treatment groups. The expression levels of FAM26F and the cytokines (IL-7, IL-11, IL-15, and IL-27) were evaluated using Real-time qPCR in PBMCs of treatment groups. Results revealed significant downregulation of IL-7 and IL-27 in infected individuals compared to healthy controls, persisting even after treatment. This suggests the crucial roles of these immune modulators in facilitating the necessary T-cell response for viral clearance. IL-11 expression also remained suppressed post-treatment, supporting viral clearance by restoring the Th1 response. The decrease in IL-11 levels during treatment indicates the restoration of the Th1 response, vital for viral clearance. IL-15, the key cytokine regulating cytotoxic cells (NKT and NK cells), displayed consistent expression across all sample groups, indicating maintained IL-15-induced cytotoxicity in both control and infected individuals. Additionally, FAM26F expression was reduced in the HCV-infected group compared to controls, but higher in HCV-recovered cases, potentially due to reduced infection and enhanced immunity. In conclusion, this research unveils the relationship between FAM26F and HCV infection, highlighting the virus's tendency to suppress cytokine and FAM26F expression. An effective treatment strategy for establishing an ideal host immune response may involve restoring FAM26F and cytokine expression to their normal levels.

## 1. Introduction

Since the Hepatitis C virus (HCV) was first identified as the cause of non-A non-B hepatitis in 1989, it has continued to represent a threat to global health. Significant gaps in our knowledge

---

**Data Availability Statement:** The datasets used and analysed during the current study are Uploaded as supplementary information.

**Funding:** This research received support from both the National University of Science & Technology and the Institute of Biomedical Genetic Engineering (IBGE). The study was funded by NUST through the PhD student's research funds. The funders had no role in study design, data collection and analysis, decision to publish, or preparation of the manuscript.

**Competing interests:** The authors have declared that no competing interests exist.

and the efficient treatment of this viral infection remain after decades of research and medical improvements [1]. A significant majority of individuals (between 55% and 85%) develop chronic hepatitis C during the acute phase of HCV infection, which normally lasts for around six months.

A relentless struggle persists between viral persistence and the components of the immune system, potentially leading to a series of severe liver conditions, spanning from hepatitis to fibrosis, cirrhosis, and, in the direst scenarios, hepatocellular carcinoma (HCC).These terrible results highlight the urgent need for better preventative and treatment methods [2].

With seven recognized genotypes and more than 80 subgroups, the HCV virus has an astounding genetic variety that presents a major obstacle in the development of a vaccine. To address the viral diversity and provide widespread protection, this astonishing variance poses a severe barrier to vaccine development [3–5]. Two oral Direct Acting Antivirals (DAAs), Boceprevir and Telaprevir, were approved in 2011, significantly altering the landscape of HCV treatment. This represented a dramatic shift from the previous reliance on ribavirin (RBV) and conventional and pegylated interferon therapy [6].

Although we have made tremendous progress in understanding the significance of adaptive immune responses in preventing viral infections, especially in the context of HIV and HCV, there is still a clear knowledge gap regarding innate immune responses. This shortcoming highlights the value of future research into the intricate workings of the innate immune system and its possible roles in HCV management and clearance [7, 8]. It is understood that cell-mediated immunity, which is predominantly controlled by T cells, is a crucial weapon in the struggle against HCV [9]. Even though they cause liver damage, CD8 cells play a critical role in the elimination of HCV-infected hepatocytes, highlighting the delicate balance needed for efficient viral management while minimising injury [10].

The immune response to HCV infection hinges on maintaining a delicate equilibrium between an intense immune reaction, which may suppress the infection (albeit with a risk of unspecific inflammation), and a limited inflammatory response that paves the way for chronicity [9]. Potential immune regulators in this complex interplay encompass Natural Killer (NK) cells, regulatory T cells (Treg), Tumor Growth Factor (TGF) β, and Interleukin (IL-10) [11]. Additionally, IL-7 governs the activity of T follicular helper (Tfh) cells in individuals with chronic HCV, IL-15 holds the potential to impede HCV replication via the ERK pathway, and IL-27 has been associated with antiviral effects. Consequently, these molecules may serve as potential therapeutic agents in the battle against chronic HCV [12–14].

Family with sequence similarity 26, member F (FAM26F), a component whose possible contributions to viral infections has drawn attention but are still poorly understood, is one of the intriguing players in the fight against HCV [15]. Prior research has documented the involvement of these genes in viral clearance across various studies. However conflicting or no clear rolehas been reported for the potential role of these immune markers for HCV clearance in response to various treatment regimes. Our objective was to gather data on the responses elicited by these markers in both untreated and diverse treatment groups. This information will serve as a foundation for extending our investigations to identify potential therapeutic targets. Similarly, additional research is necessary to fully understand the roles that cytokines like IL-15 and IL-11 play in inflammation and infection in order to fully realize their therapeutic potential. Furthermore, promising targets for intervention include newly discovered immune response regulators against HCV, such as interleukin-10 (IL-10), regulatory T cells (Treg), tumour growth factor (TGF), and natural killer (NK) cells. It would be wise to investigate their functions and possibilities for modulation in HCV management.

Our study aims to investigate how the presence of HCV infection affects the expression of four crucial cytokines (IL-7, IL-11, IL-15, and IL-27) as well as the differential expression of

FAM26F. We also examine the post-treatment changes in these cytokines in chronic HCV patients receiving Sofosbuvir, pegylated interferon, or standard interferon. In order to provide insight on potential variances in immune responses and regulatory elements that may eventually affect treatment outcomes, our goal is to understand how each treatment regimen affects the immune system. This research is an important first step in elucidating the complex network of interactions between host regulatory immune factors (IL-7, IL-11, IL-15, IL-27, and FAM26F) and HCV, opening the door to more effective approaches to address this major global health issue. We are well aware of the urgent need to address the unresolved issues and knowledge gaps around HCV infection and treatment as we set out on this adventure.

## 2. Materials and methods

### 2.1 Ethical statement

NUST Departmental Ethical Committee and Centre for Liver and Digestive Diseases, Holy Family Hospital, Rawalpindi and Khan Research Laboratories Hospital, Islamabad approved the study. The enrolled patients with legal guardians provided assent for the participation in study, and they were assured that the data would be kept confidential. The samples were collected between 1st January, 2015 and 25th February, 2018.

### 2.2 Study subjects

A pool of 399 chronic HCV patient blood samples for this study collected from General Teaching Hospital and Holy Family Hospital, Pakistan, with confirmed genotype 3. The samples were collected between 2015 and 2018, and included patients with standard IFN/ribavirin treatment (SIT) or pegylated IFN/ribavirin (PEG), or Sofosbuvir (SOFO) treatment. Blood samples for HCV treatment naïve (TN) patients were also collected. Patients with autoimmune or alcoholic liver disease or co-infection were excluded. Demographic and basic clinical attributes of the patients are mentioned in Table 1.

Patients were divided into 13 analysis groups (Fig 1) based on treatment stage and treatment options. The study included TN HCV patients, those at rapid/early virological response (RVR/EVR), end of treatment (ETR), and sustained virological response (SVR). Blood samples from 28 healthy individuals were collected as control samples. Only 244 samples were checked for gene expression, resulting in 15–20 individuals per analysis group due to stipulated limitations (Fig 1).

### 2.3 Host immune regulatory genes

Five regulatory immune factors were selected for this study: FAM26F, IL7, IL11, IL15 andIL27 based on the role they play for regulation of both the innate and the adaptive immune system.

**Table 1. Demographic and basic clinical attributes of the enlisted patients.**

| Variable | Total | Responders | Non-responders | P value* |
|---|---|---|---|---|
| | (n = 368) | (n = 241) | (n = 127) | |
| Gender (Male/Female) | | 98/143 | 47/80 | 0.496 |
| Baseline viral load range (IU/ml) | | $2 \times 10^3$-$1 \times 10^7$ | $1 \times 10^3$-$2 \times 10^7$ | |
| Age, year (average ± SD) | | 39 ± 15 | 38 ± 16 | 0.459 |
| ALT, IU/L (average ± SD) | | 76± 77 | 103 ± 89 | **<0.001** |

(* P values < 0.05 were considered significant)

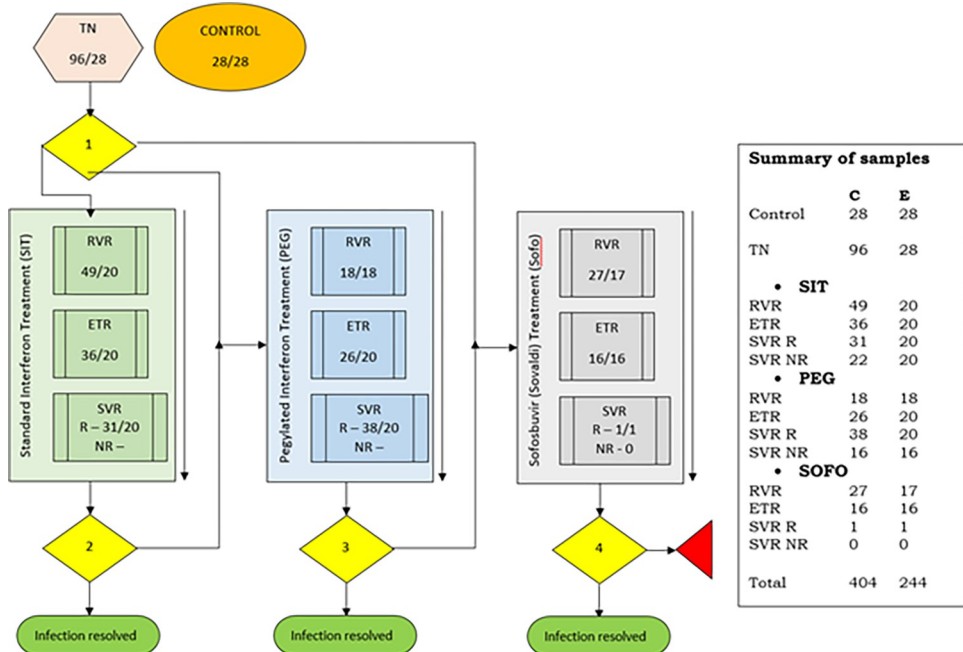

**Fig 1. Patient analysis groups for expression studies: TN, treatment naïve; RVR, rapid virological response; ETR, end of treatment response; SVR, sustained virological response; R, responders; NR, non-responders; the number under treatment stage represent the number of samples collected (C)/used for expression studies (E).** Arrows right to treatment box represent direction of treatment while the small boxes within the treatment box indicate the number of samples collected for each treatment response group. Yellow diamonds indicate: 1. Consultant decides what treatment regime to follow; 2. SVR for SIT achieved then infection resolved otherwise move to PEG; 3. SVR for PEG achieved then infection resolved otherwise move to Sofo; 4. SVR for Sofo achieved then infection resolved otherwise infection remains unresolved.

## 2.4 RNA extraction and purity

For QRT-PCR, RNA was extracted from Peripheral Blood Mononuclear Cells (PBMCs) using the Trizol method [16]. Before quantifying RNA was treated with 10 μl of 10X DNAse buffer to prevent any contamination from DNA. Its quality was checked prior to synthesis of cDNA by running onbleach gel stained with ethidium bromide (EtBr) [17]. Distinct 28S and 18S rRNA bands were visualized under UV for intact total RNA in DigiGenius gel documentation system (Syngene, United Kingdom). Furthermore, the extracted RNA was quantified within 2 hours of elution on a NanoDrop 2000 UV-Vis Spectrophotometer (Thermo Scientific). A 260/280 ratio of nearly 1.8 to 2 indicated RNA purity.

## 2.5 cDNA synthesis

Moloney Murine Leukemia Virus Reverse Transcriptase (M-MLV RT) (Invitrogen, Cat No: 28025013) was used for cDNA synthesis. Experiments were performed following the respective manufacturer's instructions. Finally, the cDNA was diluted in the ratio 1:10 for further downstream experiments. The synthesis of cDNA was confirmed by GAPDH PCR.

## 2.6 QRT-PCR optimization

The RNA level of each gene was quantified through real-time QRT-PCR using DyNAmo HS SYBR Green QPCR kit (#F-410L; Thermo Scientific, CA, USA). Primer pairs for the target genes were designed using primer3 software [18], which were then optimized by gradient PCR

**Table 2. Primers for host immune regulatory genes.**

| No. | Gene | Sequence | Product size | | Tm |
|---|---|---|---|---|---|
| | | | RNA | Genome | |
| 1 | GAPDH F | CCTGCACCACCAACTGCTTA | 74 | none | 60 |
| | GAPDH R | CATGAGTCCTTCCACGATACCA | | | |
| 2 | FAM26F F | TGTTGGGCTGGATCTTGATAG | 98 | none | 60 |
| | FAM26F R | CTGCTGCTTCCTGTTCCAA | | | |
| 3 | IL7 F | CGGATTAGGGCATTTGAGAA | 168 | 168 | 60 |
| | IL7 R | GCAACTGATACCTTACATGGATTG | | | |
| 4 | IL11 F | AGCTGCAAGGTCAAGATGGT | 159 | 159 | 60 |
| | IL11 R | TCCTTAGCCTCCCTGAATGA | | | |
| 5 | IL15 F | TGGATGCAAAGAATGTGAGG | 182 | 182 | 60 |
| | IL15 R | TTGAAATGCCGAGTGTTTTG | | | |
| 6 | IL27 F | GAGCAGCTCCCTGATGTTTC | 154 | 1718 | 60 |
| | IL27 R | AGCTGCATCCTCTCCATGTT | | | |

to determine their optimal annealing temperature. Primer sequences for the selected target genes and GAPDH as the house keeping gene are given in Table 2. ABI PRISM 7000 Sequence Detection System (Applied Biosystems, California USA) was used to conduct the Real-time qPCR. mRNA levels were calculated as copy number relative to 100 copies of GAPDH. The relativistic expression (rE) of target gene was determined as:

$$rE = 100 \text{ x } 2-\Delta Ct$$

Where, $\Delta Ct$ = mean Ct (target gene) minus mean Ct (GAPDH)

## 2.7 Statistical analysis

Statistical analysis was carried out using Graph-Pad Prism 6.0 software (Graph-Pad Software, San Diego, CA USA) where $p < 0.05$ was considered significant. Unpaired t-test at 90% confidence interval was used to compare the sample groups where Gaussian approximation was not assumed.

## 3. Results

Differential expression of FAM26F and four important cytokines (IL-7, IL-11, IL-15, and IL-27)were assessed in 244 chronic HCV patients and 28 controls who were receiving standard interferon, pegylated interferon, and Direct Acting Antivirals (DAAs) treatment. Examined expression in chronic HCV patients at rapid viral response (RVR), end of treatment response (ETR), and sustained virological responses (SVR), comparing with healthy controls.

### 3.1 Differential expression of FAM26F

No significant difference was observed between control and TN individuals. For SIT, an increase in expression was observed in individuals who achieved SVR R compared to control/TN, whereas similar levels were observed for SVR NR and control/TN. FAM expression was slightly increased in SVR R compared to SVR NR (p value: 0.0349). Results indicated that there was statistically not significant change in expression at RVR from TN while the levels decreased for ETR and then increased for SVR R and SVR NR (Fig 2A). In case of PEG, FAM26F a trend of upregulated expression was observed in RVR, ETR and SVR R, while downregulated trend in SVR NR compared to control (p value: 0.0351), indicating that a trend

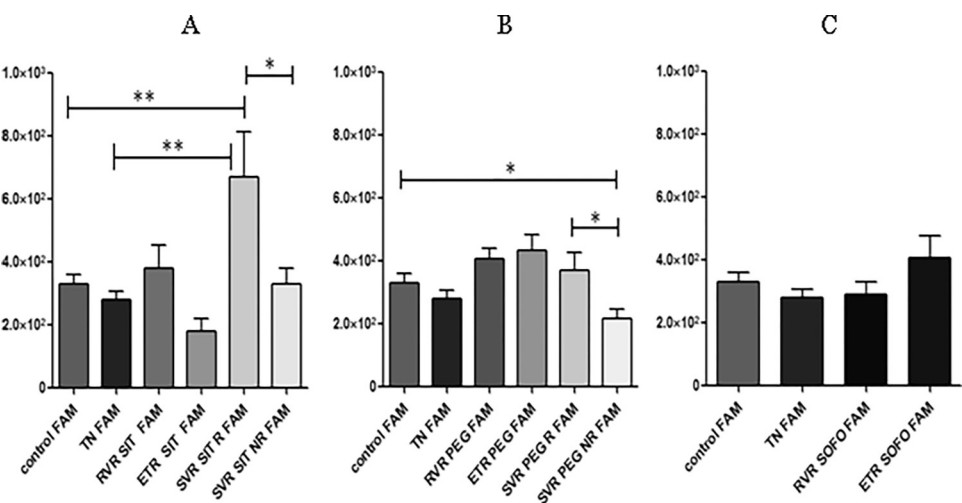

**Fig 2.** Differential expression of *FAM26F*, across (A) SIT, (B) PEG and (C) SOFO treatment. All the experiments were performed in triplicates (±SD) and the significance was calculated by Student's t-test(*P < 0.01, **P < 0.01, ***P < 0.001). Y axis displays relative GAPDH expression, x axis represents sample groups.

of down regulation was seen in FAM expression in SVR NR than SVR R individuals as shown in Fig 2B.

For SOFO there was no noticeable difference observed between control/TN, RVR and ETR individuals as shown in Fig 2C.

As far as differential expression of FAM at RVR, ETR and SVR R/NR across SIT, PEG and SOFO is concerned; expression at RVR for SIT and PEG was similar but slightly decreased for SOFO. At ETR the expression for SIT was much lower than either PEG or SOFO. Levels at SVR R/NR for SIT were higher than PEG as depicted in Fig 3A–3D but all trends were not found statistical significant.

### 3.2 Differential expression of IL7

Significant down regulation in TN IL7 expression (p value: 0.0024) was observed between treatment and control individuals. For SIT, a trend of down regulation was noted for treated individuals compared to TN, while similar expression for SVR R and SVR NR was seen. Change in expression in RVR from TN was detected that was statistically not significant, whereas ETR and SVR R/NR levels were downregulated as depicted in (Fig 4A).

PEG treatment shows upregulation of IL7 levels in RVR; downregulation at ETR, upregulation at SVR R and SVR NR. Significant upregulation in IL7 expression was seen between SVR R and SVR NR (p value: 0.0012). SVR NR and SVR R showed similar IL7 expression in PEG, while SOFO levels were significantly downregulated at RVR compared to ETR. Furthermore, remarkable difference between TN and RVR individuals was observed while not in TN and ETR individuals for SOFO as shown in (Fig 4B and 4C).

IL7 differential expression at RVR, ETR and SVR R/NR across SIT, PEG and SOFO showed significant difference between RVR SOFO and RVR SIT/PEG but no difference between RVR SIT and RVR PEG. At ETR the expression for SIT was similar to SOFO while higher than at PEG. Finally IL7 levels at SVR R/NR for SIT were lower than for PEG as depicted in (Fig 5A–5D).

### 3.3 Differential expression of IL-11

No significant difference was observed between control and TN individuals. For SIT, the IL11 levels remained same for RVR and TN/control but significantly decreased for ETR and further

                                                                 

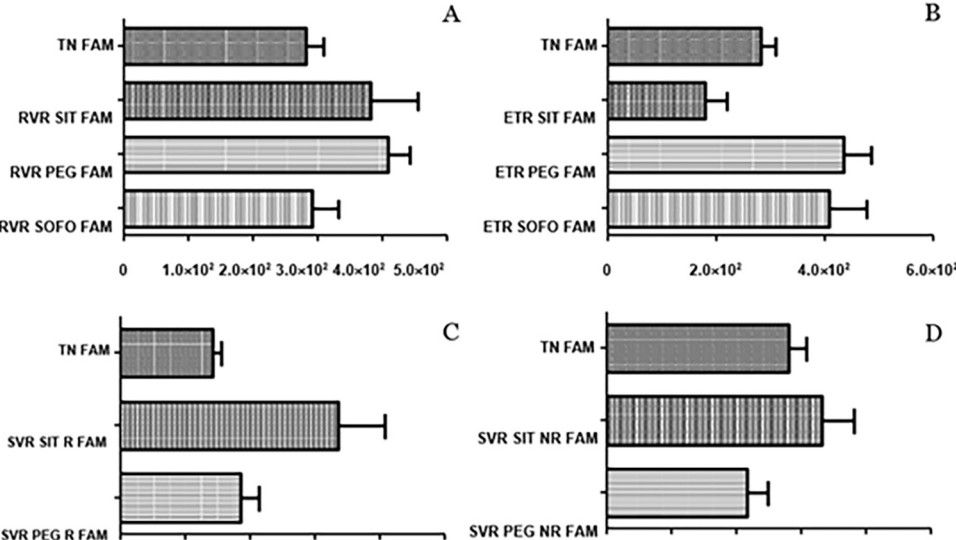

**Fig 3.** Differential expression of *FAM26F* across SIT, PEG and SOFO treatment at (A) RVR, (B) ETR, (C) SVR R and (D) SVR NR compared with TN. All the experiments were performed in triplicates (±SD) and the significance was calculated by Student's t-test(*P < 0.05, **P < 0.01, ***P < 0.001).

decreased for SVR R/NR. No significant difference in IL11 expression was observed between SVR R and SVR NR (p value: 0.2224). Compared to control/TN IL11 expression was notably decreased in individuals at SVR R/NR (Fig 6A).

In case of PEG, the levels at RVR and ETR were markedly lower compared to control/TN but slightly increased for SVR R/NR compared to ETR. Therefore, contrary to SIT where the IL11 expression decreased up to SVR, in case of PEG, IL11 expression decreases till ETR slightly increases for SVR. However for SIT and PEG, levels under treatment were significantly lower than control/TN. For SOFO as well RVR and ETR levels were significantly lower than control/TN, indicating under treatment IL11 expression decreased (Fig 6B and 6C).

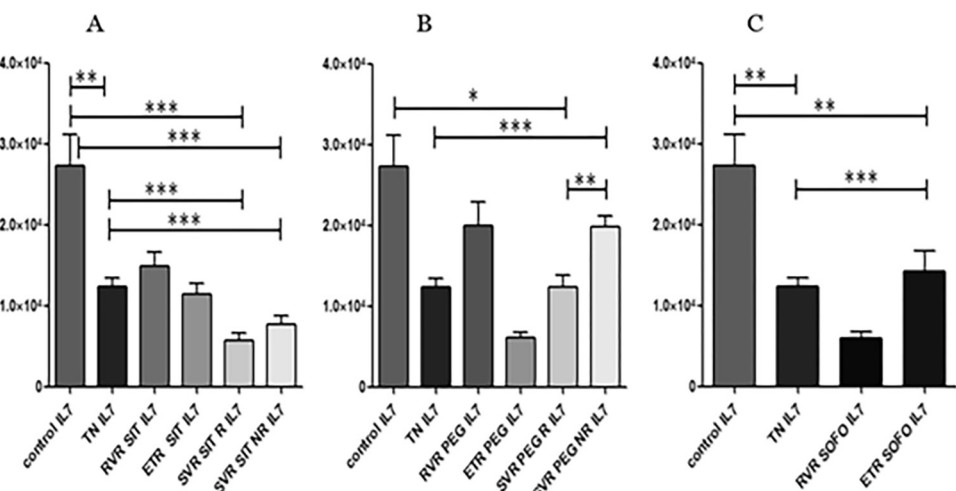

**Fig 4.** Differential expression of *IL7*, across (A) SIT, (B) PEG and (C) SOFO treatment. All the experiments were performed in triplicates (±SD) and the significance was calculated by Student's t-test(*P < 0.05, **P < 0.01, ***P < 0.001). X axis indicate the different sample groups and Y axis show the relative expression values per GAPDH.

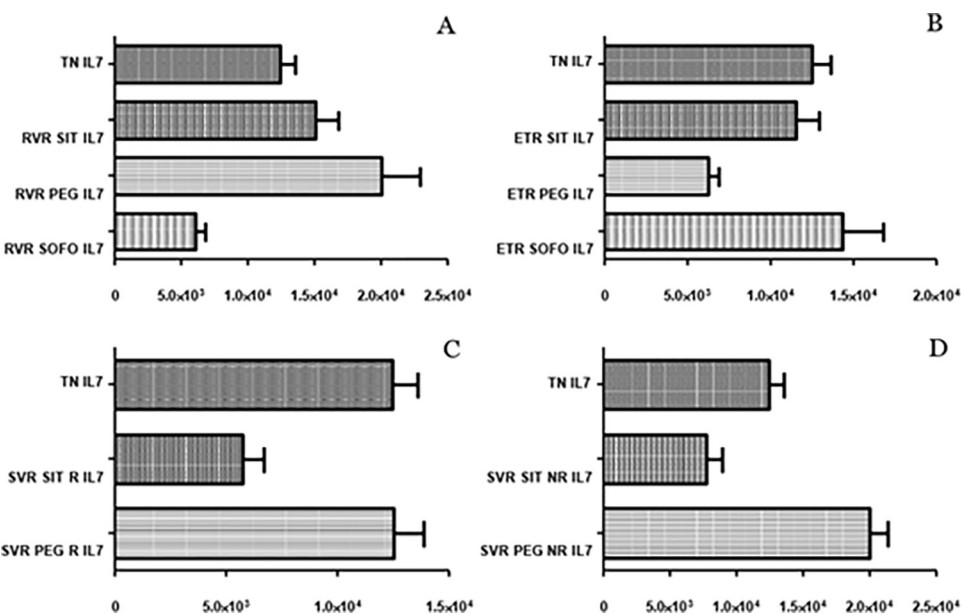

**Fig 5.** Differential expression of IL7 across SIT, PEG and SOFO treatment at RVR (A), ETR (B), SVR R (C) and SVR NR (D) compared with TN. All the experiments were performed in triplicates (±SD) and the significance was calculated by Student's t-test(*P < 0.05, **P < 0.01, ***P < 0.001).

The study found that IL11 expression in SIT, PEG, and SOFO RVR for SIT was significantly upregulated than RVR PEG or SOFO, while downregulated at RVR SOFO. Furthermore, at ETR expression pattern was found same for SIT and SOFO, and SVR R/NR compared to PEG and SOFO, with PEG having significantly higher levels than SIT (p value: 0.0008) as depicted in (Fig 7A–7D).

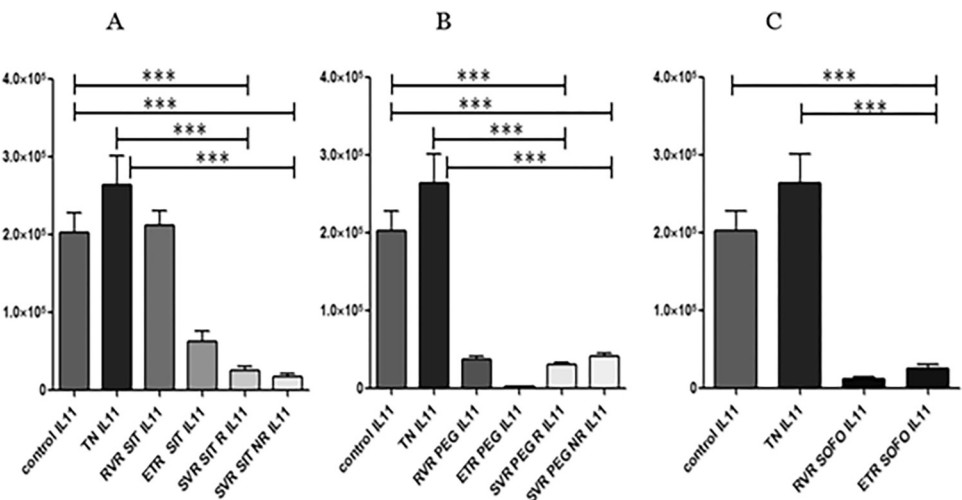

**Fig 6.** Differential expression of *IL-11*, across (A) SIT, (B) PEG and (C) SOFO treatment. All the experiments were performed in triplicates (±SD) and the significance was calculated by Student's t-test(*P < 0.05, **P < 0.01, ***P < 0.001). Y axis show the relative expression values per GAPDH and the X axis indicate the different sample groups.

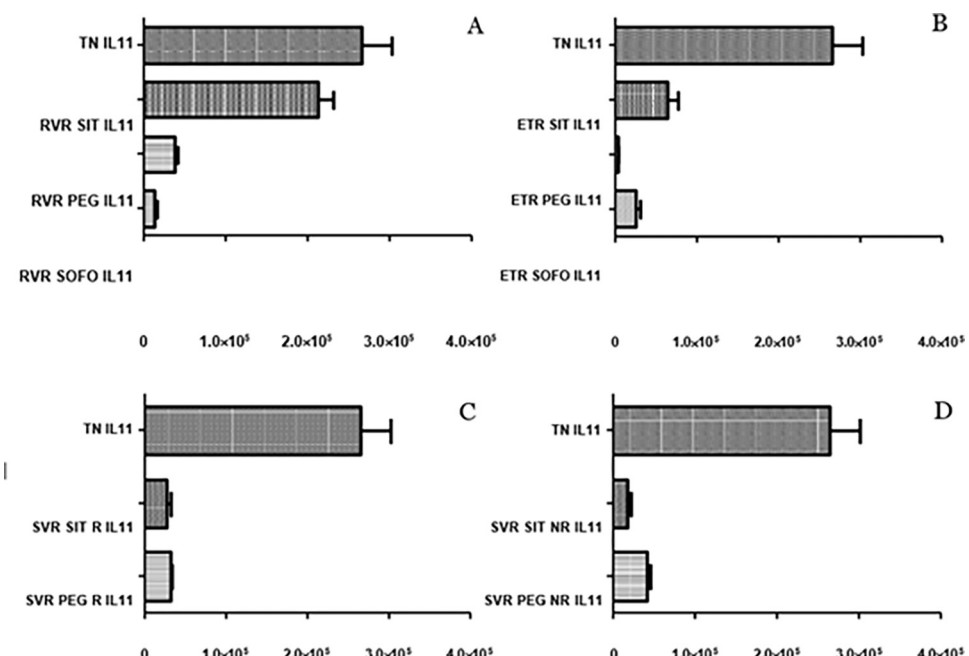

**Fig 7.** Differential expression of *IL-11* across SIT, PEG and SOFO treatment at (A) RVR, (B) ETR, (C) SVR R and (D) SVR NR compared with TN. All the experiments were performed in triplicates (±SD) and the significance was calculated by Student's t-test(*P < 0.05, **P < 0.01, ***P < 0.001).

## 3.4 Differential expression of IL-15

No significant difference in the expression of IL15 was observed between control and TN individuals. For SIT, IL15 expression remained almost the same for RVR, ETR and SVR NR compared to TN/control but significantly decreased for SVR R compared to TN/control (p value: 0.0014/0.0028). No marked difference in IL15 expression was observed between SVR R and SVR NR (p value: 0.0783). Therefore, IL15 expression remained consistent before and after SIT treatment as shown in (Fig 8A).

In case of PEG, IL15 levels at RVR were higher than TN/control and then decreased for ETR. Levels for SVR R were markedly higher compared to ETR while levels for SVR NR were lower compared to ETR. The IL15 levels for SVR R for PEG were significantly higher than SVR NR, contrary to SIT where the IL15 differential expression at SVR R and NR were similar. For SIT and PEG alike, levels under treatment were more or less similar to control/TN. For SOFO, the levels at RVR and ETR were lower than TN/control (p value: 0.0021 and 0.5979) (Fig 8B and 8C).

IL-15 expression in SIT, PEG, and SOFO was analyzed, showing similar expression at RVR for SIT to TN; lower compared to PEG, and higher compared to SOFO. ETR levels were similar to TN, while SVR R for SIT was downregulated compared to PEG and TN and SVR NR expression was downregulated for PEG compared to SIT but barely significant (p value: 0.0526) as depicted in (Fig 9A–9D).

## 3.5 Differential expression of IL-27

IL27 differential expression in control was significantly higher than in TN, meaning HCV infection caused a decrease in IL27 expression (<0.0001). However, the levels at RVR and ETR for SIT had no significant difference from TN individuals. For SIT, the IL27 expression

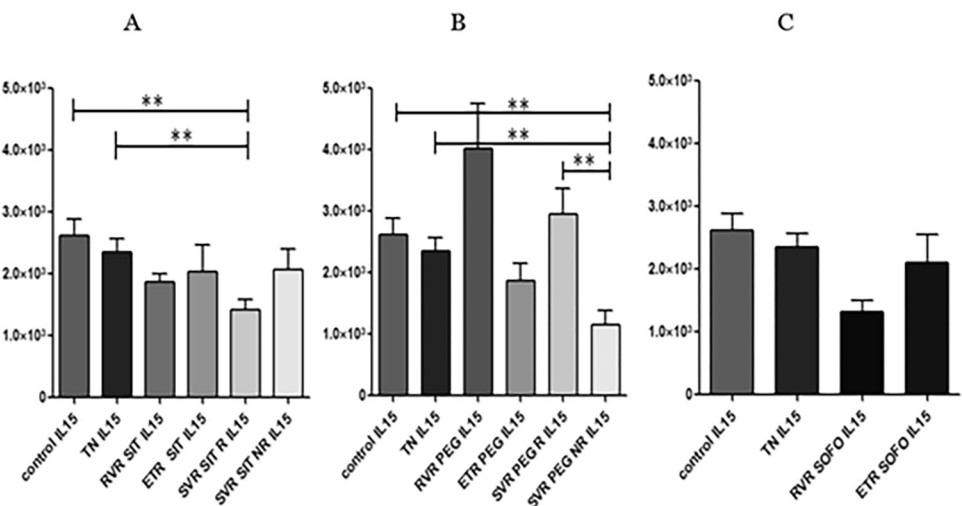

**Fig 8.** Differential expression of *IL-15* across (A) SIT, (B) PEG and (C) SOFO treatment. All the experiments were performed in triplicates (±SD) and the significance was calculated by Student's t-test(*P < 0.05, **P < 0.01, ***P < 0.001). Y axis show the relative expression values per GAPDH and the X axis indicate the different sample groups.

significantly dropped for SVR R from ETR but increased again for SVR NR. Finally the levels at SVR R SIT were slightly lower than SVR NR SIT (p value: 0.0188). Overall, compared to the control all infected individuals with or without treatment had considerably lower IL27 expression as depicted in Fig 10A.

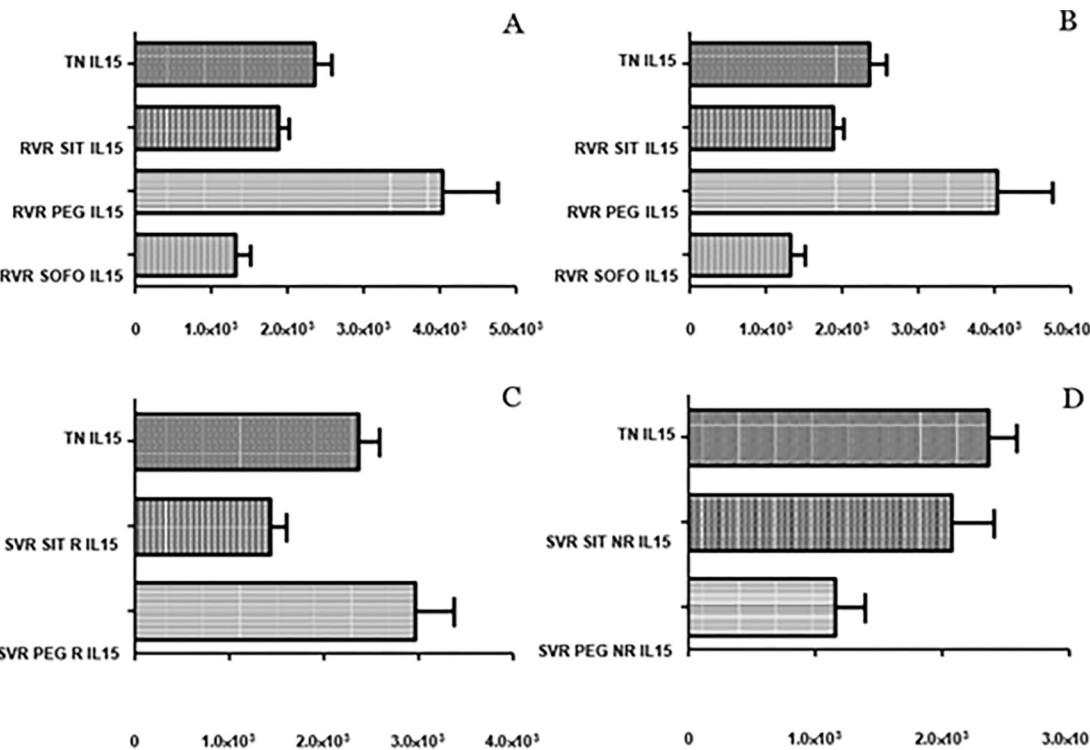

**Fig 9.** Differential expression of *IL15* across SIT, PEG and SOFO treatment at RVR (A), ETR (B), SVR R (C) and SVR NR (D) compared with TN. All the experiments were performed in triplicates (±SD) and the significance was calculated by Student's t-test (*P < 0.05, **P < 0.01, ***P < 0.001).

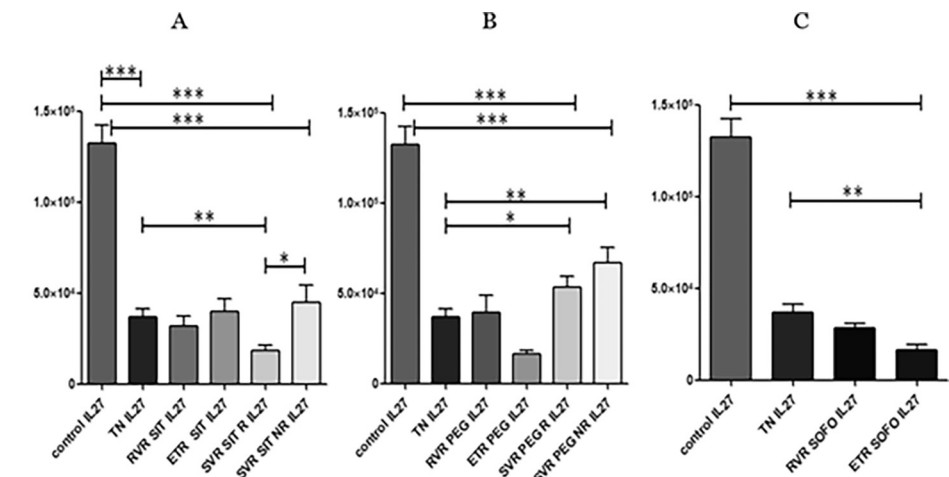

**Fig 10.** Differential expression of *IL-27* across (A) SIT, (B) PEG and (C) SOFO treatment. All the experiments were performed in triplicates (±SD) and the significance was calculated by Student's t-test(*P < 0.05, **P < 0.01, ***P < 0.001). Y axis show the relative expression values per GAPDH and the X axis indicate the different sample groups.

In case of PEG, IL27 levels at RVR, ETR and SVR R/NR were markedly lower compared to control, however RVR and ETR were similar and lower to TN, respectively. IL27 levels were significantly increased for SVR R/NR compared to ETR. Therefore, contrary to SIT where the IL27 expression was more or less similar for RVR, ETR and SVR, in case of PEG, IL27 expression for ETR decreased and then increased for SVR R/NR. Difference between SVR R and NR was not significant. However for SIT and PEG, levels for the infected were significantly lower than control. For SOFO also RVR levels were slightly higher than ETR but lower than TN and significantly lower than control, indicating that under infection IL27 expression diminishes as shown in (Fig 10B and 10C).

The study found no significant difference in IL-27 expression across SIT, PEG, and SOFO in RVR individuals. ETR SIT expression was similar to TN but significantly upregulated than PEG or SOFO. Expression at SVR R for SIT were significantly downregulated than PEG (p value: <0.0001), whereas the expression for SVR NR for SIT and PEG had no significant difference as depicted in (Fig 11A–11D). A comprehensive table detailing p values for the various comparisons made for all immune factors assessed is given in Table 3. All data sets underlying the findings reported are given in S1 Data.

## 4. Discussion

Cytokines play a pivotal role as regulators of the immune system, and understanding their modulation during HCV infection is vital in unravelling the pathogenesis of HCV and the development of chronic infection. While much of the previous research on cytokines in the context of HCV pathogenesis has centred around inflammation, with a focus on both inflammatory and anti-inflammatory cytokines, our study takes a distinct approach by delving into their trends in expression within different treatment groups [19, 20].

Cytokines exhibit remarkable plasticity in their ability to stimulate immune cells and orchestrate immune responses, often categorized as inflammatory, anti-inflammatory, or regulatory. In this study, we place particular emphasis on their immune regulatory functions, employing mRNA levels as a more cost-effective means of assessment compared to protein-based measurements [21]. Our investigation involved a thorough examination of the

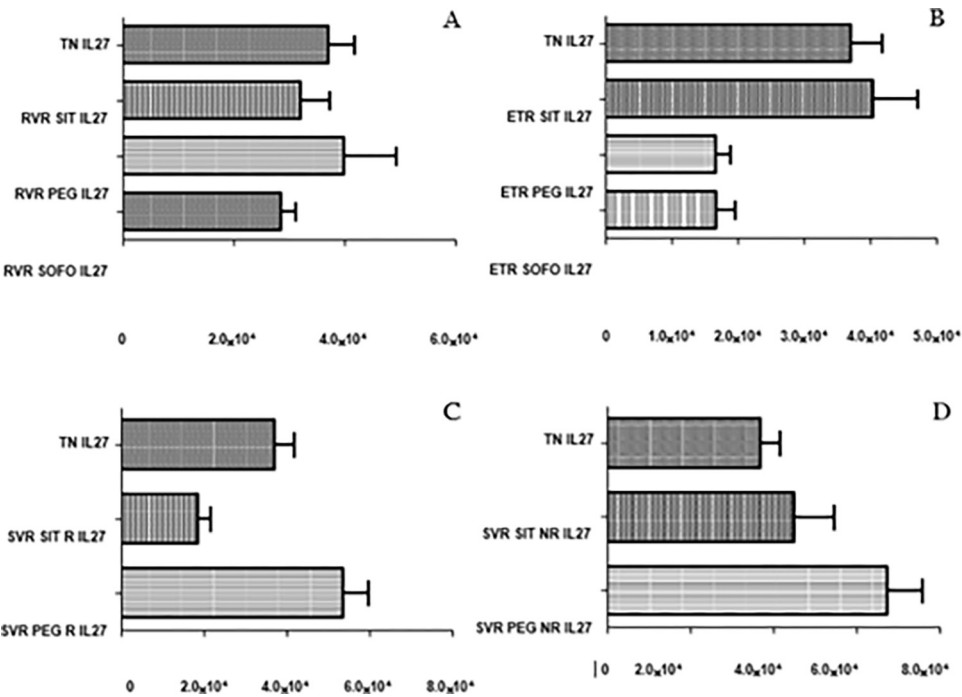

**Fig 11.** Differential expression of *IL-27* across SIT, PEG and SOFO treatment at (A) RVR, (B) ETR, (C) SVR R and (D) SVR NR compared with TN. All the experiments were performed in triplicates (±SD) and the significance was calculated by Student's t-test(*P < 0.05, **P < 0.01, ***P < 0.001).

differential expression of interleukins, namely IL7, IL11, IL15, and IL27, in PBMCs from both treatment-naive individuals and those who had received standard interferon, pegylated interferon, or Direct Acting Antivirals (DAAs) treatment. Our findings unveiled significant decreases in the expression of IL7 and IL27 among infected individuals. This downregulation of cytokines is a concerning factor, as it can lead to a weakened T-cellular response, which is crucial for the clearance of the virus. A robust T-cell response is essential for effective viral control.

IL7 is the key cytokine associated with the pre B- and pre T-cell growth. It regulates lymphopoiesis and is essential for T-cell production, survival and expansion in the periphery and most importantly the development of T-cell memory. IL7 is especially important for the regular T-cell production and development but not so for B-cell however the lymphopaenia do result in a weakened B-cell response [12, 22].

Here, the differential expression of IL7 was checked in the PBMCs of HCV chronic patients: TN and under SIT, PEG and SOFO at different treatment stages along with healthy controls. Some very interesting observations were made; first the expression levels of IL7 for infected TN patients were severely depleted (p value: 0.0024) compared to the controls and secondly the IL7 levels remained below par even after treatment. However patients under SOFO made the most marked recovery in IL7 levels compared to SIT and PEG. Levels for PEG were slightly higher than those for SIT. This indicates that during chronic HCV infection IL7 levels drop which would cause a reduced T-cell response. Especially the ability of the memory T-cells to combat infection recurrence will be severely hampered leading to not only relapse after treatment but the development of persistent chronic HCV infection [23, 24]. The inability of the treatment to re-establish IL7 levels is also intriguing and indicates the inability of resident immune cells to produce IL7 required bringing all T-cell populations–helper, cytotoxic and

**Table 3. The p values for the sample groups.**

| # | Sample groups analyzed | p value summary | Is p value significant? (p < 0.05) | p value |
|---|---|---|---|---|
| 1 | control FAM vs TN FAM | Ns | No | 0.2551 |
| **2** | **control FAM vs SVR SIT R FAM** | ** | **yes** | **0.0079** |
| 3 | control FAM vs SVR SIT NR FAM | Ns | no | 0.9748 |
| **4** | **TN FAM vs SVR SIT R FAM** | ** | **yes** | **0.0036** |
| 5 | TN FAM vs SVR SIT NR FAM | Ns | no | 0.3434 |
| **6** | **SVR SIT R FAM vs SVR SIT NR FAM** | * | **yes** | **0.0349** |
| 7 | TN FAM vs RVR SIT FAM | Ns | no | 0.186 |
| **8** | **TN FAM vs ETR SIT FAM** | * | **yes** | **0.0405** |
| **9** | **RVR SIT FAM vs ETR SIT FAM** | * | **yes** | **0.0482** |
| **10** | **ETR SIT FAM vs SVR SIT R FAM** | ** | **yes** | **0.0074** |
| **11** | **ETR SIT FAM vs SVR SIT NR FAM** | * | **yes** | **0.0316** |
| 12 | control FAM vs SVR PEG R FAM | Ns | no | 0.5129 |
| **13** | **control FAM vs SVR PEG NR FAM** | * | **yes** | **0.0245** |
| 14 | TN FAM vs SVR PEG R FAM | Ns | no | 0.1408 |
| 15 | TN FAM vs SVR PEG NR FAM | Ns | no | 0.138 |
| **16** | **SVR PEG R FAM vs SVR PEG NR FAM** | * | **yes** | **0.0351** |
| **17** | **TN FAM vs RVR PEG FAM** | ** | **yes** | **0.0098** |
| **18** | **TN FAM vs ETR PEG FAM** | ** | **yes** | **0.0066** |
| 19 | RVR PEG FAM vs ETR PEG FAM | Ns | no | 0.6705 |
| 20 | ETR PEG FAM vs SVR PEG R FAM | Ns | no | 0.4239 |
| **21** | **ETR PEG FAM vs SVR PEG NR FAM** | ** | **yes** | **0.0012** |
| 22 | TN FAM vs RVR SOFO FAM | Ns | no | 0.8439 |
| 23 | TN FAM vs ETR SOFO FAM | Ns | no | 0.0552 |
| 24 | RVR SOFO FAM vs ETR SOFO FAM | Ns | no | 0.141 |
| 25 | control IL7 vs TN IL7 | ** | yes | 0.0024 |
| 26 | control IL7 vs SVR SIT R IL7 | *** | yes | <0.0001 |
| 27 | control IL7 vs SVR SIT NR IL7 | *** | yes | 0.0001 |
| 28 | TN IL7 vs SVR SIT R IL7 | *** | yes | <0.0001 |
| 29 | TN IL7 vs SVR SIT NR IL7 | *** | yes | <0.0001 |
| 30 | SVR SIT R IL7 vs SVR SIT NR IL7 | Ns | no | 0.1999 |
| 31 | TN IL7 vs RVR SIT IL7 | Ns | no | 0.2045 |
| 32 | TN IL7 vs ETR SIT IL7 | Ns | no | 0.6037 |
| 33 | RVR SIT IL7 vs ETR SIT IL7 | Ns | no | 0.1367 |
| 34 | ETR SIT IL7 vs SVR SIT R IL7 | ** | yes | 0.0019 |
| 35 | ETR SIT IL7 vs SVR SIT NR IL7 | Ns | no | 0.0527 |
| 36 | control IL7 vs SVR PEG R IL7 | * | yes | 0.0104 |
| 37 | control IL7 vs SVR PEG NR IL7 | Ns | no | 0.1553 |
| 38 | TN IL7 vs SVR PEG R IL7 | Ns | no | 0.9703 |
| 39 | TN IL7 vs SVR PEG NR IL7 | *** | yes | 0.0003 |
| 40 | SVR PEG R IL7 vs SVR PEG NR IL7 | ** | yes | 0.0012 |
| 41 | TN IL7 vs RVR PEG IL7 | * | yes | 0.0103 |
| 42 | TN IL7 vs ETR PEG IL7 | *** | yes | 0.0004 |
| 43 | RVR PEG IL7 vs ETR PEG IL7 | *** | yes | 0.0002 |
| 44 | ETR PEG IL7 vs SVR PEG R IL7 | *** | yes | 0.0006 |
| 45 | ETR PEG IL7 vs SVR PEG NR IL7 | *** | yes | <0.0001 |
| 46 | TN IL7 vs RVR SOFO IL7 | Ns | no | 0.4764 |
| 47 | TN IL7 vs ETR SOFO IL7 | *** | yes | 0.0002 |

*(Continued)*

**Table 3.** (Continued)

| # | Sample groups analyzed | p value summary | Is p value significant? (p < 0.05) | p value |
|---|---|---|---|---|
| 48 | RVR SOFO IL7 vs ETR SOFO IL7 | ** | yes | 0.0074 |
| 49 | control IL11 vs TN IL11 | Ns | no | 0.2138 |
| 50 | control IL11 vs SVR SIT R IL11 | *** | yes | <0.0001 |
| 51 | control IL11 vs SVR SIT NR IL11 | *** | yes | <0.0001 |
| 52 | TN IL11 vs SVR SIT R IL11 | *** | yes | <0.0001 |
| 53 | TN IL11 vs SVR SIT NR IL11 | *** | yes | <0.0001 |
| 54 | SVR SIT R IL11 vs SVR SIT NR IL11 | Ns | no | 0.2224 |
| 55 | TN IL11 vs RVR SIT IL11 | Ns | no | 0.2331 |
| 56 | TN IL11 vs ETR SIT IL11 | *** | yes | <0.0001 |
| 57 | RVR SIT IL11 vs ETR SIT IL11 | *** | yes | <0.0001 |
| 58 | ETR SIT IL11 vs SVR SIT R IL11 | * | yes | 0.0425 |
| 59 | ETR SIT IL11 vs SVR SIT NR IL11 | * | yes | 0.0137 |
| 60 | control IL11 vs SVR PEG R IL11 | *** | yes | <0.0001 |
| 61 | control IL11 vs SVR PEG NR IL11 | *** | yes | <0.0001 |
| 62 | TN IL11 vs SVR PEG R IL11 | *** | yes | <0.0001 |
| 63 | TN IL11 vs SVR PEG NR IL11 | *** | yes | <0.0001 |
| 64 | SVR PEG R IL11 vs SVR PEG NR IL11 | Ns | no | 0.0521 |
| 65 | TN IL11 vs RVR PEG IL11 | *** | yes | <0.0001 |
| 66 | TN IL11 vs ETR PEG IL11 | *** | yes | <0.0001 |
| 67 | RVR PEG IL11 vs ETR PEG IL11 | *** | yes | <0.0001 |
| 68 | ETR PEG IL11 vs SVR PEG R IL11 | *** | yes | <0.0001 |
| 69 | ETR PEG IL11 vs SVR PEG NR IL11 | *** | yes | <0.0001 |
| 70 | TN IL11 vs RVR SOFO IL11 | *** | yes | <0.0001 |
| 71 | TN IL11 vs ETR SOFO IL11 | *** | yes | <0.0001 |
| 72 | RVR SOFO IL11 vs ETR SOFO IL11 | * | yes | 0.0474 |
| 73 | control IL15 vs TN IL15 | Ns | no | 0.4622 |
| 74 | control IL15 vs SVR SIT R IL15 | ** | yes | 0.0014 |
| 75 | control IL15 vs SVR SIT NR IL15 | Ns | no | 0.2158 |
| 76 | TN IL15 vs SVR SIT R IL15 | ** | yes | 0.0028 |
| 77 | TN IL15 vs SVR SIT NR IL15 | Ns | no | 0.4677 |
| 78 | SVR SIT R IL15 vs SVR SIT NR IL15 | Ns | no | 0.0783 |
| 79 | TN IL15 vs RVR SIT IL15 | Ns | no | 0.0947 |
| 80 | TN IL15 vs ETR SIT IL15 | Ns | no | 0.4965 |
| 81 | RVR SIT IL15 vs ETR SIT IL15 | Ns | no | 0.7031 |
| 82 | ETR SIT IL15 vs SVR SIT R IL15 | Ns | no | 0.1751 |
| 83 | ETR SIT IL15 vs SVR SIT NR IL15 | Ns | no | 0.9588 |
| 84 | control IL15 vs SVR PEG R IL15 | Ns | no | 0.4784 |
| 85 | control IL15 vs SVR PEG NR IL15 | ** | yes | 0.0026 |
| 86 | TN IL15 vs SVR PEG R IL15 | Ns | no | 0.1688 |
| 87 | TN IL15 vs SVR PEG NR IL15 | ** | yes | 0.0027 |
| 88 | SVR PEG R IL15 vs SVR PEG NR IL15 | ** | yes | 0.0023 |
| 89 | TN IL15 vs RVR PEG IL15 | * | yes | 0.0135 |
| 90 | TN IL15 vs ETR PEG IL15 | Ns | no | 0.1898 |
| 91 | RVR PEG IL15 vs ETR PEG IL15 | * | yes | 0.0138 |
| 92 | ETR PEG IL15 vs SVR PEG R IL15 | * | yes | 0.0425 |
| 93 | ETR PEG IL15 vs SVR PEG NR IL15 | Ns | no | 0.0744 |
| 94 | TN IL15 vs RVR SOFO IL15 | ** | yes | 0.0021 |

(*Continued*)

**Table 3.** (Continued)

| # | Sample groups analyzed | p value summary | Is p value significant? (p < 0.05) | p value |
|---|---|---|---|---|
| 95 | TN IL15 vs ETR SOFO IL15 | Ns | no | 0.5979 |
| 96 | RVR SOFO IL15 vs ETR SOFO IL15 | Ns | no | 0.1154 |
| 97 | control IL27 vs TN IL27 | *** | yes | <0.0001 |
| 98 | control IL27 vs SVR SIT R IL27 | *** | yes | <0.0001 |
| 99 | control IL27 vs SVR SIT NR IL27 | *** | yes | <0.0001 |
| 100 | TN IL27 vs SVR SIT R IL27 | ** | yes | 0.0039 |
| 101 | TN IL27 vs SVR SIT NR IL27 | Ns | no | 0.4591 |
| 102 | SVR SIT R IL27 vs SVR SIT NR IL27 | * | yes | 0.0188 |
| 103 | TN IL27 vs RVR SIT IL27 | Ns | no | 0.5 |
| 104 | TN IL27 vs ETR SIT IL27 | Ns | no | 0.6996 |
| 105 | RVR SIT IL27 vs ETR SIT IL27 | Ns | no | 0.3581 |
| 106 | ETR SIT IL27 vs SVR SIT R IL27 | * | yes | 0.0108 |
| 107 | ETR SIT IL27 vs SVR SIT NR IL27 | Ns | no | 0.6821 |
| 108 | control IL27 vs SVR PEG R IL27 | *** | yes | < 0.0001 |
| 109 | control IL27 vs SVR PEG NR IL27 | *** | yes | < 0.0001 |
| 110 | TN IL27 vs SVR PEG R IL27 | * | yes | 0.0462 |
| 111 | TN IL27 vs SVR PEG NR IL27 | ** | yes | 0.0046 |
| 112 | SVR PEG R IL27 vs SVR PEG NR IL27 | Ns | no | 0.1969 |
| 113 | TN IL27 vs RVR PEG IL27 | Ns | no | 0.7845 |
| 114 | TN IL27 vs ETR PEG IL27 | ** | yes | 0.0019 |
| 115 | RVR PEG IL27 vs ETR PEG IL27 | * | yes | 0.034 |
| 116 | ETR PEG IL27 vs SVR PEG R IL27 | *** | yes | <0.0001 |
| 117 | ETR PEG IL27 vs SVR PEG NR IL27 | *** | yes | <0.0001 |
| 118 | TN IL27 vs RVR SOFO IL27 | Ns | no | 0.188 |
| 119 | TN IL27 vs ETR SOFO IL27 | ** | yes | 0.0036 |
| 120 | RVR SOFO IL27 vs ETR SOFO IL27 | ** | yes | 0.0078 |
| 121 | RVR SIT FAM vs RVR PEG FAM | Ns | no | 0.7883 |
| 122 | RVR SIT FAM vs RVR SOFO FAM | Ns | no | 0.3198 |
| 123 | RVR PEG FAM vs RVR SOFO FAM | * | yes | 0.0487 |
| 124 | ETR SIT FAM vs ETR PEG FAM | *** | yes | 0.0009 |
| 125 | ETR SIT FAM vs ETR SOFO FAM | * | yes | 0.011 |
| 126 | ETR PEG FAM vs ETR SOFO FAM | Ns | no | 0.764 |
| 127 | SVR SIT R FAM vs SVR PEG R FAM | Ns | no | 0.051 |
| 128 | SVR SIT NR FAM vs SVR PEG NR FAM | Ns | no | 0.0659 |
| 129 | RVR SIT IL7 vs RVR PEG IL7 | Ns | no | 0.1338 |
| 130 | RVR SIT IL7 vs RVR SOFO IL7 | *** | yes | 0.0002 |
| 131 | RVR PEG IL7 vs RVR SOFO IL7 | *** | yes | < 0.0001 |
| 132 | ETR SIT IL7 vs ETR PEG IL7 | ** | yes | 0.0031 |
| 133 | ETR SIT IL7 vs ETR SOFO IL7 | Ns | no | 0.3693 |
| 134 | ETR PEG IL7 vs ETR SOFO IL7 | * | yes | 0.0112 |
| 135 | SVR SIT R IL7 vs SVR PEG R IL7 | *** | yes | 0.0004 |
| 136 | SVR SIT NR IL7 vs SVR PEG NR IL7 | *** | yes | < 0.0001 |
| 137 | RVR SIT IL11 vs RVR PEG IL11 | *** | yes | < 0.0001 |
| 138 | RVR SIT IL11 vs RVR SOFO IL11 | *** | yes | < 0.0001 |
| 139 | RVR PEG IL11 vs RVR SOFO IL11 | *** | yes | < 0.0001 |
| 140 | ETR SIT IL11 vs ETR PEG IL11 | ** | yes | 0.0041 |
| 141 | ETR SIT IL11 vs ETR SOFO IL11 | * | yes | 0.0139 |

(*Continued*)

**Table 3.** (Continued)

| # | Sample groups analyzed | p value summary | Is p value significant? (p < 0.05) | p value |
|---|---|---|---|---|
| 142 | ETR PEG IL11 vs ETR SOFO IL11 | ** | yes | 0.0076 |
| 143 | SVR SIT R IL11 vs SVR PEG R IL11 | Ns | no | 0.3738 |
| 144 | SVR SIT NR IL11 vs SVR PEG NR IL11 | *** | yes | 0.0008 |
| 145 | RVR SIT IL15 vs RVR PEG IL15 | ** | yes | 0.0033 |
| 146 | RVR SIT IL15 vs RVR SOFO IL15 | * | yes | 0.0279 |
| 147 | RVR PEG IL15 vs RVR SOFO IL15 | *** | yes | 0.0007 |
| 148 | ETR SIT IL15 vs ETR PEG IL15 | Ns | no | 0.7521 |
| 149 | ETR SIT IL15 vs ETR SOFO IL15 | Ns | no | 0.9153 |
| 150 | ETR PEG IL15 vs ETR SOFO IL15 | Ns | no | 0.6691 |
| 151 | SVR SIT R IL15 vs SVR PEG R IL15 | *** | yes | 0.0008 |
| 152 | SVR SIT NR IL15 vs SVR PEG NR IL15 | Ns | no | 0.0526 |
| 153 | RVR SIT IL27 vs RVR PEG IL27 | Ns | no | 0.4701 |
| 154 | RVR SIT IL27 vs RVR SOFO IL27 | Ns | no | 0.6108 |
| 155 | RVR PEG IL27 vs RVR SOFO IL27 | Ns | no | 0.3143 |
| 156 | ETR SIT IL27 vs ETR PEG IL27 | ** | yes | 0.009 |
| 157 | ETR SIT IL27 vs ETR SOFO IL27 | * | yes | 0.0138 |
| 158 | ETR PEG IL27 vs ETR SOFO IL27 | Ns | no | 0.9965 |
| 159 | SVR SIT R IL27 vs SVR PEG R IL27 | *** | yes | <0.0001 |
| 160 | SVR SIT NR IL27 vs SVR PEG NR IL27 | Ns | no | 0.0972 |

The p values for the sample groups analysed using unpaired t-test at 90% confidence interval (Ns—not significant). Significant values are given in bold.

memory–in homeostasis. Meaning that after HCV builds chronic infection, levels of IL7 plummet and is not recovered even after treatment and since IL7 remains central to T-cell response, patients fail to mount proper cytotoxic T-cell response [25, 26]. On the other hand the observation that patients under SOFO have higher IL7 levels indicate that contrary to interferons the inhibition of viral activity by direct attack is more efficient in restoration of IL7 levels hence T-cell homeostasis. Nevertheless it is important to further investigate and correlate these observations with cellular studies, measuring the different cellular populations at different treatment stages and linking them to IL7 levels. Furthermore analysis of T-cell response after induction of exogenous IL7 should also be checked to determine if and how IL7 is able to improve T-cell response.

Prior investigations have linked IL-11 expression to viral infections, a correlation that our study corroborates. IL-11 and its corresponding receptors exhibit expression in a range of tissues, prominently including the liver [27]. It has been demonstrated that IL-11 administration can reduce inflammatory responses in a range of chronic inflammatory diseases, sepsis induced by lipopolysaccharide, macrophage inflammation, nephrotoxic nephritis, and T-cell-mediated liver injury. The observed decrease in IL-11 levels (p value: 0.0008) in our study signifies elevated inflammation following HCV infection, which corroborates our previous findings.

Treatment with standard interferon (SIT), pegylated interferon (PEG), or Sofosbuvir (SOFO) initiated downregulation of IL-11 expression, with levels in control individuals and those receiving TN (treatment-naive) being comparable. This modulation of IL-11 levels during therapy suggests the restoration of the Th1 response, a major cellular immunity mechanism vital for viral clearance.IL-15, known as the primary cytokine regulating cytotoxic cells such as NKT and NK cells, as well as a unique class of T-cells known as γδ T-cells, exhibited

steady and comparable expression levels across all sample groups [28, 29]. This suggests that IL-15-induced cytotoxicity is maintained both in control individuals and those infected with HCV. Interestingly, it has been observed that chronic HCV infections can result in aberrant, non-specific NK cell activity, which undergoes a shift and becomes more targeted upon the initiation of treatment [30, 31]. Notably, IL-15 levels were relatively higher in individuals who achieved End of Treatment Response (ETR) and Sustained Virological Response with Relapse (SVR R), indicating the potential significance of IL-15 levels in the specific NK cell response.

Various viruses induce the expression of IL-15, and its role in antiviral responses varies from virus to virus [32]. In the context of HCV, low circulating IL-15 levels have been linked to high viremia and poor disease outcomes, which is in line with our results. A decrease in IL-15 levels suggests impaired IL-15 production, potentially affecting virus control. The production of IL-15 is influenced by mature dendritic cells (DCs) and NK cells, with interferon-alpha (IFN-α) serving as a crucial stimulus. Reduction in IL-15 levels indicates a diminished antiviral response against viral clearance, mirroring observations in both HBV and HCV infections [33].

Vast body of literature suggests a potential role of IL27 as a very important cytokine in the regulation adaptive immune response, especially that mediated by the T cells. It triggers clonal expansion of naïve CD4+ T-cells and stimulates CD4+ cells to produce IFN-γ [34]. IL27 has been indicated to confer long term survival of the T-cell population through regulating Tregs. IL27 is also the key cytokine inducing Th follicular cells to produce IL21, which is central to the proliferation of multiple immune cells like T, B, and NK cells [35]. Studies with IL27 knockout mice (IL27RA−/−) showed severely diminished production of high affinity antibodies and inflammation with very few IFN-γ producing T and Th17 cells [36]. It has also been implicated that IL27 may have only a minor role for infection clearance but is essential in down-regulating inflammation during chronic phases of the infection, reducing the intensity of inflammation associated tissue injury. This anti-inflammatory regulatory role of IL27 is established via IL10 production from the various T-cell populations including Tregs. A number of studies also indicate that IL27 suppresses Th2 and Th17 cells [37]. In fibroblasts IL27 induces IL18 binding protein a natural IL18 antagonist which also regulates inflammation through limiting IL18 activities. In T & B-cells IL27 over expression directly down regulates IL6R signalling producing an anti-inflammatory effect [38]. So potentially IL27 is an adaptive immunity regulatory through effecting T-cell function. Still its versatility is obvious as a regulator since it enhances or silences inflammatory response by various mechanisms that are not fully understood.

In present investigation it was observed that the IL27 expression was severely down regulated for the HCV infected group than the healthy controls. Moreover during the course of treatment for any option (SIT, PEG or SOFO) the IL27 levels further plummeted indicating that on one side the infection caused a severe down regulation of IL27 and on the other the treatment further added to this suppression. The expression levels were slightly but not significantly restored for the SVR R/NR. As far as the levels across the treatment options is concerned there was a marked increase in SVR R for PEG compared to SIT indicating that SVR following PEG had better cytokine expression than for SIT. Since IL27 is a potent immune regulatory cytokine of the adaptive immunity, down regulation in case of HCV infection suggests that viral pathogenesis flourishes and favours inflammatory environment [39]. Yet it is imperative to note that this inflammation becomes a burden for the host instead of eliminating the infection.

Direct-acting antivirals (DAAs) have been found to provide superior responses compared to interferon therapy. Specifically, pegylated interferon (PEG) has shown better outcomes than standard interferon (SIT). The modulation of host immune factors examined in this study underscores these distinctions, with different patterns of expression observed between SIT and

PEG at similar points in the treatment course. The higher documented relapse rates for SIT compared to PEG, along with the underlying molecular mechanisms discussed, could be pivotal factors influencing these outcomes.

In addition to cytokines, our study also explored the expression of FAM26F, which was found to be downregulated in HCV-infected patients compared to uninfected controls. HCV employs strategies to evade the host immune system, resulting in reduced FAM26F expression, as well as the interruption of IFN-inducing cascades. FAM26F expression is dependent on TICAM-1 and IRF-3 activation and is responsive to multiple pathways involved in the antiviral response, highlighting its potential role in the host's defense mechanisms against viral infections.

It is noteworthy that, in resource-constrained settings like Pakistan, where the majority of patients are still treated with SIT, treatment side effects and relapse post-treatment are common. This study serves as an initial step in unravelling the differences in immune responses associated with various treatment options. Such investigations are crucial in understanding the intricate relationship between HCV infection and the immune response. Cytokines, pivotal regulators of both innate and adaptive immunity, offer a window into how HCV conceals its pathogenesis to establish chronic infection. While previous research has predominantly focused on inflammation, particularly concerning inflammatory or anti-inflammatory cytokines, our approach shifts the spotlight toward the immunoregulatory aspects of these molecules. Categorizing cytokines as purely inflammatory, anti-inflammatory, or regulatory is a simplification, as their functionalities often overlap, showcasing extensive plasticity in stimulating diverse immune cells and eliciting varied immune responses. In this study, we emphasize their immune regulatory functions, employing mRNA levels for a more cost-effective and efficient assessment compared to protein-based measurements.

The study provides insight into the importance of the stated cytokines and their role in HCV pathogenesis and disease progression. However, the data needs to be validated in a larger study cohort with more patients and control samples in each group. Furthermore, the qRT-PCR results should be validated at the protein level using techniques such as ELISA or western blot.

## 5. Conclusion

In conclusion, our study sheds light on the intricate dynamics of the immune system's efforts to restore diminished T-cell and NK cellular activity following treatment, with Sofosbuvir and peg-interferon appearing to be particularly effective in this regard. The modulation of these host regulatory variables emerges as a crucial determinant in the battle against HCV infection. These findings underscore the multifaceted interplay between treatment regimens and the immune system's response, providing valuable insights into the factors influencing treatment efficacy. As we navigate the complex landscape of HCV management, further research in this direction is warranted to refine therapeutic strategies and enhance treatment outcomes.

## Supporting information

**S1 Data. All data sets underlying the findings reported in submitted manuscript are provided in supplementary data.**
(XLSX)

## Acknowledgments

We are thankful to Holy Family Hospital, Rawalpindi for providing blood samples. This research was funded by NUST student funds of Dr. Aneela Javed. The results described in this paper were part of student thesis.

## Author Contributions

**Conceptualization:** Sajid Mansoor, Aneela Javed.

**Data curation:** Sajid Mansoor, Ali Zalan, Momina Ejaz.

**Formal analysis:** Madiha Khlaid, Sajid Mansoor, Ali Zalan, Momina Ejaz, Atika Mansoor.

**Investigation:** Kehkshan Jabeen, Madiha Khlaid, Sajid Mansoor, Ali Zalan.

**Methodology:** Sajid Mansoor.

**Project administration:** Aneela Javed.

**Resources:** Aneela Javed.

**Software:** Atika Mansoor.

**Supervision:** Atika Mansoor, Aneela Javed.

**Validation:** Kehkshan Jabeen, Madiha Khlaid.

**Visualization:** Kehkshan Jabeen, Momina Ejaz.

**Writing – original draft:** Kehkshan Jabeen, Madiha Khlaid, Ali Zalan, Momina Ejaz.

**Writing – review & editing:** Kehkshan Jabeen, Madiha Khlaid, Aneela Javed.

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
