## [Decision Letter · Decision Letter 0]

12 Dec 2023

PGPH-D-23-02024

Host Immune Players and their Response to Hepatitis C Therapies

Dear Dr. Javed,

Thank you for submitting your manuscript to PLOS Global Public Health. After careful consideration, we feel that it has merit but does not fully meet PLOS Global Public Health’s publication criteria as it currently stands. Therefore, we invite you to submit a revised version of the manuscript that addresses the points raised during the review process.

We look forward to receiving your revised manuscript.

Kind regards,

Jianhong Zhou

Staff Editor

Journal Requirements:

1. In the online submission form, you indicated that "The datasets used and analysed during the current study are available from the corresponding author on reasonable request.". All PLOS journals now require all data underlying the findings described in their manuscript to be freely available to other researchers, either 1. In a public repository, 2. Within the manuscript itself, or 3. Uploaded as supplementary information.

Additional Editor Comments (if provided):

Reviewers' comments:

Reviewer's Responses to Questions

**Comments to the Author**

1. Does this manuscript meet PLOS Global Public Health’s publication criteria? Is the manuscript technically sound, and do the data support the conclusions? The manuscript must describe methodologically and ethically rigorous research with conclusions that are appropriately drawn based on the data presented.

Reviewer #1: Partly

Reviewer #2: Yes

2. Has the statistical analysis been performed appropriately and rigorously?

Reviewer #1: No

Reviewer #2: I don't know

3. Have the authors made all data underlying the findings in their manuscript fully available (please refer to the Data Availability Statement at the start of the manuscript PDF file)?

Reviewer #1: No

Reviewer #2: Yes

4. Is the manuscript presented in an intelligible fashion and written in standard English?

Reviewer #1: No

Reviewer #2: Yes

5. Review Comments to the Author

Reviewer #1: Jabeen et al. looked at the RNA expression of four genes coding for cytokines and one gene coding for a pore-forming subunit of a voltage-gated ion channel in a cohort of HCV-infected individuals and some control participants. They investigated the differential expression of those five genes across different treatments received by the HCV-infected individuals. The analytical strategy proposed is sound; however, this study has several major problems regarding the clarity of the figures, the statistical inference used, and the quality of text that I hope the authors can address. Below are the requested edits:

Major corrections:

Lines 13-15. The sentences "IL-11 expression... inhibiting the Th1 response" and "The decrease in IL-11.... restoration of Th1 response..." seem to suggest, on the one hand, that low IL-11 is associated with inhibition of Th1 response and, on the other hand, that low IL-11 is important for the Th1 response restoration. Which one is it?

Lines 71-73. There needs to be a clear justification for why the authors look at those five genes and not others. Not wanting to limit the genes queried to inflammatory markers is fine. Still, the reason why the authors selected those specific genes should be presented in the introduction or result section.

Lines 141-143: If you are not assuming a Gaussian distribution, you should use a non-parametric test like a Wilcoxon rank-sum test over a Student t-test (a t-test is a parametric test and assumes that the data is normally distributed).

Line 143: There is no correction for multiple testing anywhere in the paper.

Line 144: There is no description of the clinical characteristics of the cohort (e.g., sex, age of the participants) other than the number of participants by treatment group (presented in Figure 1). A table with those clinical parameters stratified by treatment groups should be given as Table 1.

Lines 145-146: "Investigations explores...and immune system functions." what does this sentence mean?

Lines 147, 268, 271: looking at the RNA expression of cytokines does *not* mean looking at the "immune regulatory activity" of those cytokines; do not overinterpret your work.

Line 148: Indicate when the samples were collected after initiation of the treatments.

Lines 157, 176: It is not "insignificant"; it is "not statistically significant".

Line 159: The sentence "In case of PEG, FAM26F expression was upregulated in RVR, ETR" is not supported by stats, so you should indicate that this is only a trend.

Line 163: Cite only Fig2C, there is no SOFO in Fig 2B.

Line 166-171: Mention that nothing reaches statistical significance (or add the stats to Figure 3).

Line 184: "significant downregulation of RVR compared to ETR". That does not match the stats shown in Figure 4C.

Line 187-191: There is no stat indicated in Fig 5.

line 196: "downregulated significantly for ETR" that doesn't match the stats in Figure 6A.

Lines 206-207: "RVR...significantly lower than control/TN". There is no statistical support for RVR vs control/TN, according to Fig 6C.

Line 222: Do you mean "IL15"? IL11 is written in the text.

Line 230: The stats in the text do not match the stats in Fig. 8B and Fig. 8C.

Line 238: No stat is shown in Fig. 9.

Line 246: There is no statistical support for IL27 being lower in ETR compared to TN in Fig 10B.

Line 388: Add a paragraph on the limitations of the study.

All figures show individual points for each participant, not just average expressions and error bars. Replace bar plots/histograms with jitter plots. Add axis labels. Indicate in the legend what the *, **, *** correspond to for the p-value you calculated. Put the legends of the figures at the end of the document and not in the text.

Minor corrections:

Line 7: add between these "treatment" groups. 

Line 9: indicates that those genes were looked in blood in the abstract.

Line 106: replace "digit" by "number".

Lines 108-109: replace "the treatment points when blood sample were drawn" by "the number of samples collected for each treatment response group".

Lines 119, 131: Use qRT-PCR over QRT-PCR. Be consistent with the choice of capitalization across the article (ex. line 132 you use qRT-PCR).

Fig 2a: The horizontal bar for the comparison TN vs "SVR R" should go all the way to the halfway point of the SVR R bar.

Line 181: add that you are comparing RVR vs TN. Was the upregulation of IL7 significant?

Lines 225, 249: replace "than" by "compared to".

Line 259: Table 2 is useful, but it would be easier for the reader if the stats are indicated in the figures.

Reviewer #2: 1. Collect the figure ligands in a separate part after the references

2. At RNA extraction and purity method: A pretreatment of the extracted RNA with DNAase should be done in order to eliminate any DNA contamination during the extraction process especially when using Trizol for RNA extraction. Otherwise the primer design should be done so primers have to span an exon – exon junction. Did you check this point?

3. The primers for FAM26F do not have any match at primer blast.

4. What is the role of the IDO-1 primers in this study

5. You should mention the method of quantification of your genes and in your case it should quantified by relative fold change obtained by the 2−ΔΔCt method.

6. The figures for gene expression should be measured in fold of expression not in copy number as in the link according the 2−ΔΔCt equation: https://bmcgenomics.biomedcentral.com/articles/10.1186/1471-2164-11-455/figures/3

Then repeat the statistical analysis for your study.

6. PLOS authors have the option to publish the peer review history of their article (what does this mean?). If published, this will include your full peer review and any attached files.

**Do you want your identity to be public for this peer review?** For information about this choice, including consent withdrawal, please see our Privacy Policy.

Reviewer #1: **Yes: **Slim Fourati

Reviewer #2: No

---

## [Decision Letter · Decision Letter 1]

25 Mar 2024

Host Immune Players and their Response to Hepatitis C Therapies

PGPH-D-23-02024R1

Dear Dr Javed,

We are pleased to inform you that your manuscript 'Host Immune Players and their Response to Hepatitis C Therapies' has been provisionally accepted for publication in PLOS Global Public Health.

Best regards,

Pablo Hernán Sotelo Torres

Academic Editor

Reviewer Comments (if any, and for reference):

Reviewer's Responses to Questions

**Comments to the Author**

1. If the authors have adequately addressed your comments raised in a previous round of review and you feel that this manuscript is now acceptable for publication, you may indicate that here to bypass the “Comments to the Author” section, enter your conflict of interest statement in the “Confidential to Editor” section, and submit your "Accept" recommendation.

Reviewer #1: All comments have been addressed

Reviewer #2: All comments have been addressed

2. Does this manuscript meet PLOS Global Public Health’s publication criteria? Is the manuscript technically sound, and do the data support the conclusions? The manuscript must describe methodologically and ethically rigorous research with conclusions that are appropriately drawn based on the data presented.

Reviewer #1: Yes

Reviewer #2: Yes

3. Has the statistical analysis been performed appropriately and rigorously?

Reviewer #1: Yes

Reviewer #2: I don't know

4. Have the authors made all data underlying the findings in their manuscript fully available (please refer to the Data Availability Statement at the start of the manuscript PDF file)?

Reviewer #1: Yes

Reviewer #2: Yes

5. Is the manuscript presented in an intelligible fashion and written in standard English?

Reviewer #1: Yes

Reviewer #2: Yes

6. Review Comments to the Author

Reviewer #1: The authors addressed all except one of my comments/edits. The authors did not understand what correction for multiple testing meant. The multiple comparisons problem occurs when you test multiple hypotheses simultaneously (in your case, differences of 5 markers across treatment groups). Adjustments can be made to control for false-positive findings, such as using the Benjamini-Hochberg procedure.

Reviewer #2: (No Response)

7. PLOS authors have the option to publish the peer review history of their article (what does this mean?). If published, this will include your full peer review and any attached files.

**Do you want your identity to be public for this peer review?** For information about this choice, including consent withdrawal, please see our Privacy Policy.

Reviewer #1: **Yes: **Slim Fourati

Reviewer #2: No
